EMBO
Molecular Medicine

# μLED-based optical cochlear implants for spectrally selective activation of the auditory nerve

Alexander Dieter[1,2,†,‡] (iD), Eric Klein[3,†] (iD), Daniel Keppeler[1] (iD), Lukasz Jablonski[1,4] (iD), Tamas Harczos[1,4] (iD), Gerhard Hoch[1,4], Vladan Rankovic[1,4,5] (iD), Oliver Paul[3,6] (iD), Marcus Jeschke[1,4,7] (iD), Patrick Ruther[3,6,*] (iD) & Tobias Moser[1,2,4,8,9,**] (iD)

## Abstract

Electrical cochlear implants (eCIs) partially restore hearing and enable speech comprehension to more than half a million users, thereby re-connecting deaf patients to the auditory scene surrounding them. Yet, eCIs suffer from limited spectral selectivity, resulting from current spread around each electrode contact and causing poor speech recognition in the presence of background noise. Optogenetic stimulation of the auditory nerve might overcome this limitation as light can be conveniently confined in space. Here, we combined virus-mediated optogenetic manipulation of cochlear spiral ganglion neurons (SGNs) and microsystems engineering to establish acute multi-channel optical cochlear implant (oCI) stimulation in adult Mongolian gerbils. oCIs based on 16 microscale thin-film light-emitting diodes (μLEDs) evoked tonotopic activation of the auditory pathway with high spectral selectivity and modest power requirements in hearing and deaf gerbils. These results prove the feasibility of μLED-based oCIs for spectrally selective activation of the auditory nerve.

**Keywords** cochlear implant; hearing restoration; micro-LED; neural coding; optogenetics

**Subject Categories** Biotechnology & Synthetic Biology; Neuroscience

See also: **SR Kitcher & CJC Weisz** (August 2020)

## Introduction

According to the World Health Organization (Adrian *et al*, 2019), 6.1% of the human population suffers from disabling hearing loss, with an economic impact of 750 billion dollars spent on treatment development. Approximately 90% of the cases suffer from sensorineural hearing loss resulting from dysfunction of the cochlea and/or the auditory nerve. As of today, causal therapies for sensorineural hearing loss do not exist. Hence, the methods of choice for hearing restoration are hearing aids and the electrical cochlear implant (eCI). When hearing loss is profound or complete, eCIs are used to bypass dysfunctional or lost sensory hair cells and electrically stimulate SGNs. eCIs utilize the intrinsic place-frequency map (known as tonotopy) of SGNs to provide the patient with spectral information of acoustic stimuli (Zeng *et al*, 2008). However, while open speech comprehension is achieved in most of the approximately 700,000 eCI users, there is a substantial unmet clinical need for the improvement of hearing restoration. Current spread in the electro-conductive cochlear fluids limits the spatial—and thus spectral—selectivity of SGN activation by eCIs and consequently the resolution of electrical sound encoding. This ultimately compromises signal perception, especially in noisy environments (Kral *et al*, 1998; Friesen *et al*, 2001; Middlebrooks *et al*, 2005). While efforts are being undertaken toward more spectrally selective electrical SGN activation, e.g. via multipolar stimulation (Berenstein *et al*, 2008) or direct electrode-neural interfacing (Middlebrooks & Snyder, 2007), it seems that the spread of excitation remains the bottleneck of eCIs.

Recently, it has been suggested that optical stimulation, which—as opposed to electric current—can be conveniently confined in space, could activate SGNs with higher spectral selectivity. Thus, optical cochlear implants (oCIs) could increase the frequency

1 Institute for Auditory Neuroscience and InnerEarLab, University Medical Center Göttingen, Göttingen, Germany
2 Göttingen Graduate School for Neurosciences and Molecular Biosciences, University of Göttingen, Göttingen, Germany
3 Department of Microsystems Engineering (IMTEK), University of Freiburg, Freiburg, Germany
4 Auditory Neuroscience and Optogenetics Laboratory, German Primate Center, Göttingen, Germany
5 Restorative Cochlear Genomics Group, Auditory Neuroscience and Optogenetics Laboratory, German Primate Center, Göttingen, Germany
6 BrainLinks-BrainTools, Cluster of Excellence, University of Freiburg, Freiburg, Germany
7 Cognitive Hearing in Primates Group, Auditory Neuroscience and Optogenetics Laboratory, German Primate Center, Göttingen, Germany
8 Auditory Neuroscience Group, Max Planck Institute for Experimental Medicine, Göttingen, Germany
9 Cluster of Excellence "Multiscale Bioimaging: from Molecular Machines to Networks of Excitable Cells" (MBExC), University of Goettingen, Goettingen, Germany
*Corresponding author. Tel: +49 7612 037197; E-mail: ruther@imtek.de
**Corresponding author. Tel: +49 5513 963070; E-mail: tmoser@gwdg.de
†These authors contributed equally to this work
‡Present address: Synaptic Wiring Lab, Center for Molecular Neurobiology Hamburg, University Medical Center Hamburg-Eppendorf, Hamburg, Germany

resolution of artificial sound encoding (Richter *et al*, 2011; Hernandez *et al*, 2014). Two approaches have been proposed for optical activation of SGNs: (i) direct infrared neural stimulation (Richter *et al*, 2011) and (ii) optogenetics (Hernandez *et al*, 2014). While infrared stimulation has remained controversial (Verma *et al*, 2014; Thompson *et al*, 2015; Baumhoff *et al*, 2019), the use of light-gated ion channels (Channelrhodopsins, ChRs) for optogenetic SGN stimulation has been consistently reported by several laboratories (Hernandez *et al*, 2014; Duarte *et al*, 2018; Hart *et al*, 2020). Establishing optogenetic hearing restoration is a challenging, multidisciplinary task (Kleinlogel *et al*, 2020; Moser, 2015; Weiss *et al*, 2016; Richardson *et al*, 2017). Two major objectives must be met: First, optogenetics must render SGNs light sensitive and enable neural coding at good temporal precision. Second, multi-channel oCIs must allow for spectrally more selective SGN stimulation than eCIs. Temporally precise optical SGN control has been established with fast gating ChR variants (Duarte *et al*, 2018; Keppeler *et al*, 2018; Mager *et al*, 2018). Furthermore, spatially confined SGN stimulation by individual optical fibers has been shown to elicit neural responses with higher spectral precision than electrical stimulation (Hernandez *et al*, 2014; Dieter *et al*, 2019). Thus, studies on the biomedical prerequisites to enable optical sound encoding have revealed very promising results.

Toward the development of multi-channel optical cochlear implants, oCIs have been fabricated both with commercially available light-emitting diodes (LEDs) and with gallium nitride-based, custom-developed thin-film LEDs in the micrometer range (µLEDs) (Goßler *et al*, 2014; Schwaerzle *et al*, 2016; Klein *et al*, 2018; Xu *et al*, 2018). While LED-based implants (Schwaerzle *et al*, 2016; Xu *et al*, 2018) typically carried 10–15 LEDs with emitters sized from 0.22 to 1 mm, µLED-based implants were manufactured with up to 144 emitters of 60 × 60 µm (Klein *et al*, 2018). Individual µLEDs of these implants achieve an optical power output of approximately 1 mW (corresponding to an emittance of approximately 400 mW/mm² at the µLED surface), with a maximum temperature increase of 1 K (driven at 10 mA, 1 s in water) (Klein *et al*, 2018), and are thus efficient enough to drive most ChRs and safe enough for *in vivo* application. Peak intensity and light extraction of these µLEDs could further be increased (by 95% and 83%, respectively) by the implementation of conical concentrators and spherical micro-lenses on the emitter surface (Klein *et al*, 2019b). However, optogenetic SGN activation *in vivo* by any of these implants has not been demonstrated yet.

In this study, we combined viral gene transfer of the ChR-variant *CatCh* (Kleinlogel *et al*, 2011) into SGNs of adult Mongolian gerbils with optical stimulation by 16-channel, µLED-based oCIs. We demonstrate µLED-mediated activation of the spiral ganglion by 32-channel recordings of multi-neuronal clusters in the auditory midbrain in hearing and deaf gerbils in a tonotopic manner. Modest SGN activation was achieved even with some of the individual µLEDs, and the strength of responses substantially increased when recruiting additional emitters. Finally, µLED-based optogenetic stimulation, even with four neighboring µLEDs, achieved increased spectral selectivity as compared to electrical stimulation via clinical-style eCIs. This proof of increased frequency resolution by µLED-based, multi-channel oCIs raises hope that optogenetic hearing restoration might indeed overcome the major limitations of eCIs and increase the quality of artificial hearing for deaf patients in the future.

# Results

## Multi-channel optical control of the auditory nerve

To render SGNs light sensitive, we intramodiolarly injected a suspension of adeno-associated virus (AAV-PHP.B; Deverman *et al*, 2016) carrying the calcium translocating channelrhodopsin *CatCh*, fused to the reporter protein eYFP, under control of the human synapsin promoter, into the cochlea of adult gerbils (Wrobel *et al*, 2018; Dieter *et al*, 2019). Functional expression of *CatCh* was probed by recordings of auditory brainstem activity upon illumination of the cochlea with a laser-coupled optical fiber via the round window as early as 4 weeks after virus injection (Fig 1A). Opsin function (robust optically evoked auditory brainstem responses, oABRs) could be demonstrated in 15/35 animals (~43%; Fig EV1). *CatCh* expression in SGNs, and lack of obvious *CatCh* signal in inner hair cells, was demonstrated by post-mortem immunohistochemistry in a subset of oABR-positive animals (Fig 1B–D), while only very sparse opsin expression was found in an oABR-negative animal (Fig EV2). Animals with detectable oABRs have been used for subsequent electrophysiological recordings of multi-unit activity in the central nucleus of the inferior colliculus (ICC) using linear 32-channel multi-electrode arrays (IC datasets could be recorded in only 12 out of these 15 oABR-positive animals). For multi-channel optical stimulation, we used oCIs housing 16 individually addressable µLEDs (peak wavelength: 462 nm; Klein *et al*, 2018; 60 × 60 µm, Fig 1E and F) with a pitch of either 100, 150, or 250 µm (center-to-center distance) embedded in a biocompatible epoxy and medical-grade silicone (Fig 1G; Klein *et al*, 2018). The maximum power output after surface roughening amounted to 0.76 mW (± 0.21 mW SD) for individual µLEDs driven with a current of 10 mA (Fig 1H) and 3.15 mW (± 1.1 mW SD) when all emitters of an oCI were driven with a current of 40 mA (Appendix Fig S1). Using a retroauricular approach, the oCIs were inserted into the cochlea of isoflurane-anesthetized gerbils via the round window or via a cochleostomy, whereby LEDs of the implants covered maximally a third of the basilar membrane length (Fig 1I and J).

## Acoustic calibration

To interpret neural activation in response to oCI stimulation and compare data across animals, electrode positioning along the tonotopic axis of the ICC was physiologically confirmed by acoustic stimulation using pure tones of varying frequency and intensity. Frequency tuning of the recording sites has been accessed in all hearing animals after performing cochlear and cranial surgeries and measuring oABRs. Characteristic frequencies were derived for each responsive electrode, and tonotopic slopes were calculated for each animal by a linear fit of characteristic frequencies as a function of recording depth (Fig EV3A). Median tonotopic slopes amounted to 4.34 octaves/mm (± 0.42 median absolute deviation, $n = 11$), which is consistent with previous studies where tonotopic slopes were reported between 4.08 and 4.58 octaves/mm (Harris *et al*, 1997; Schnupp *et al*, 2015; Dieter *et al*, 2019). In deafened animals, the recording probe was stereotactically inserted and the median tonotopic slope of regular hearing animals was assigned since acoustic mapping could not be performed ($n = 5$; Fig EV3B).

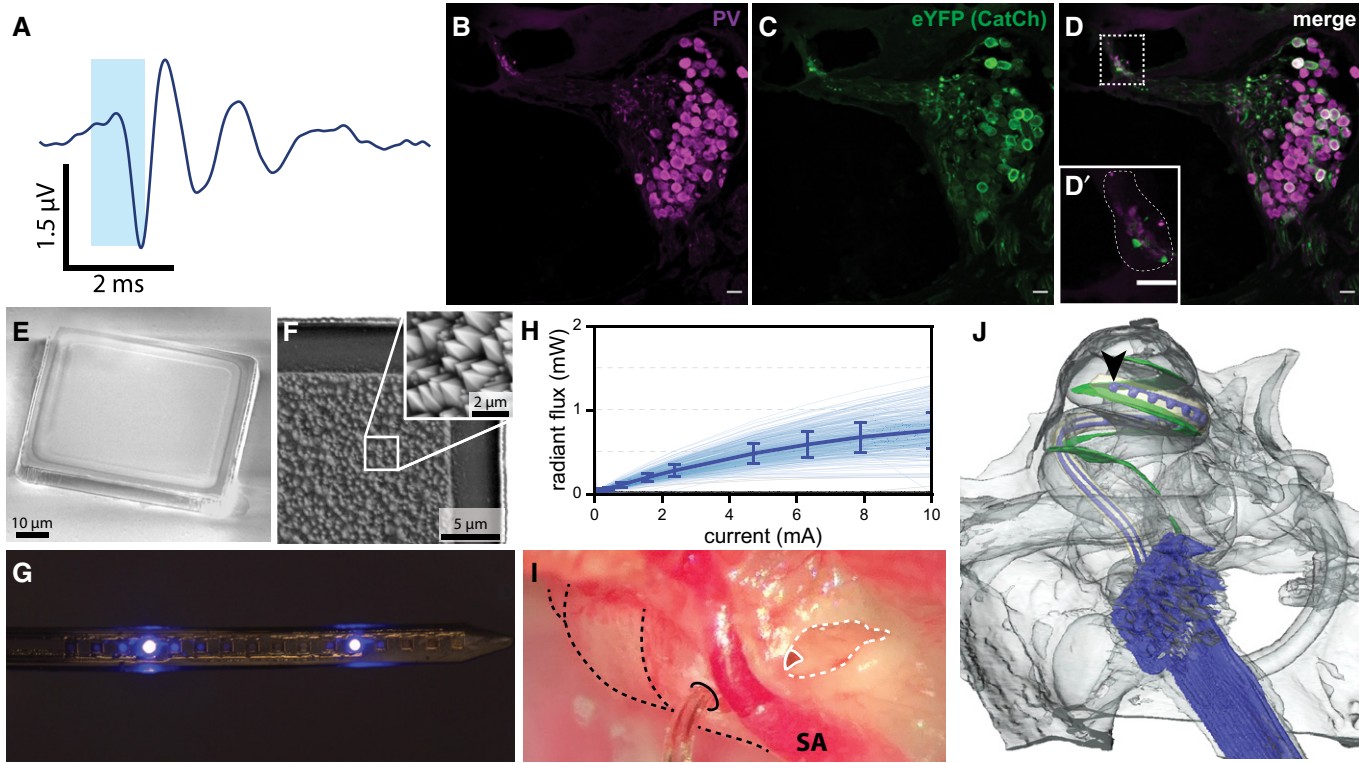

**Figure 1. µLED-based oCI.**

A   Optically evoked auditory brainstem response in an *AAV-CatCh*-injected gerbil in response to a 1 ms laser pulse of ~35 mW (mean of 1,000 stimulus presentations at 10 Hz repetition rate).

B–D   *Catch*-eYFP-expressing SGNs in the apical cochlear turn identified by parvalbumin expression (B) and *CatCh*-eYFP (C). The inlay (D′) of the merged immunostaining (D) indicates the lack of *CatCh*-EYFP signal in inner hair cells (dashed white line). Scale bar: 20 µm.

E, F   Scanning electron micrographs of a µLED (60 µm × 60 µm footprint) structured on a sapphire substrate showing the non-emitting p-contact side (E) and the emitting side of a µLED (F), transferred onto and embedded into an epoxy substrate. The GaN surface of the µLED has been roughened by KOH etching showing characteristic pyramidal structures (F, inset).

G   Picture of an oCI carrying 16 individually addressable µLEDs with a pitch of 100 µm on a flexible substrate, µLEDs #5 and #13 (from the tip) are active.

H   Radiant flux of individual µLEDs as a function of driving current. The thick line indicates the mean, error bars the SD of the mean. Non-functional µLEDs (which did not emit light, 158 out of 560 µLEDs) have been excluded.

I   oCI inserted into the cochlea (dashed black lines) via a cochleostomy in the basal cochlear turn (black, solid line). The round window niche is highlighted by a dashed white line, the round window by a solid white line. SA: stapedial artery.

J   3D X-ray tomography reconstruction of a 16-channel oCI implanted in a gerbil cochlea via the round window. Cables and µLEDs are marked in blue, and the most apical µLED is indicated by the black arrowhead; the basilar membrane is marked in green. µLEDs have a size of 60 × 60 µm.

## ICC activation with oCI stimulation

Inferior colliculus activation by oCI stimulation of SGNs was first evaluated based on peri-stimulus time histograms in response to maximum stimulus parameters (i.e., 16 active µLEDs at maximum intensity, ~3.15 mW; raw multi-unit trace, scatter plot, and PSTH are shown in Appendix Fig S2), and neural responses were found to occur 3.5–17.5 ms after stimulus onset. Since a transient stimulus artifact was observed in some traces at stimulus onset, the time window to quantify potential responses was set to 3–20 ms in order to avoid the artifact without discarding neural responses. During this time window, multi-unit activity in response to a given stimulus of increasing intensity was sorted in a two-dimensional response matrix according to the corresponding recording site and stimulus intensity. From this matrix, the cumulative discrimination index (d-prime, d′) was calculated based on spike rates in response to increasing stimulation intensities,

starting with a zero intensity condition (i.e., no stimulation, baseline; Middlebrooks & Snyder, 2007; Richter *et al*, 2011; Dieter *et al*, 2019).

The cumulative d′ quantifies the change in response strength as SD from baseline; i.e., a d′ of 1 is equal to a rise in firing rate by one SD. To estimate the patterns of ICC activation, spatial tuning curves (STCs) were constructed based on iso-contour lines at integer d′ values (Fig 2A–D). A subset of animals has been deafened by bilateral intracochlear application of kanamycin, which leads to hair cell loss and hence creates a model of sensorineural hearing loss (Wrobel *et al*, 2018). Deafening was confirmed by recordings of auditory brainstem activity in response to acoustic clicks. Mean thresholds of click-evoked auditory brainstem responses amounted to 30 ± 0 dB SPL before deafening (*n* = 5), and no responses were observed with clicks up to 100 dB SPL 5–10 days after kanamycin application (Fig 2E). Optogenetic activation of the auditory nerve was still possible in deafened animals, as demonstrated by optically

evoked ICC activity (Fig 2F and G). Neural responses in the ICC vanished after the animal was sacrificed (Fig 2H).

When stimulating SGNs with individual μLEDs of the oCI at maximum intensity, only approximately one-third of the μLEDs (206/528) evoked neural responses in the ICC (i.e., d′ values equal or greater than 1 at least at one recording site). An average maximal d′ value of 1.97 (± 0.71 SD) was obtained from responsive multi-units (Fig 2I). However, responses substantially increased when recruiting additional μLEDs: Upon maximal stimulation with four neighboring μLEDs (~0.99 ± 0.25 mW, resulting from a driving current of 10 mA), approximately two-thirds of μLED blocks (89/132) evoked neural responses in the ICC with an average maximal d′ of 2.86 (± 0.83 SD). A maximal stimulation with all 16 μLEDs of the oCI switched on resulted in an average d′ of 3.35 (± 0.65 SD, at ~3.15 mW; Fig 2I). For comparison, SGN stimulation with a laser-coupled optical fiber delivering light of ~35 mW achieved an average maximal d′ of 4.27 (± 0.52 SD). The strength of neural responses did not differ between hearing and deafened animals (Wilcoxon rank sum test, $P = 0.26/0.80/0.43/1$ for individual/block/all μLEDs and optical fiber stimulation, respectively).

Mild responses (d′ = 1.8, 2.0, and 2.5) were also observed upon driving all 16 μLEDs of 3 out of 7 oCIs in two non-injected, hearing control animals (probably due to an opto-acoustic effect; Baumhoff *et al*, 2019), but responses have never been observed in deafened control (no AAV-*CatCh* injection; $n = 7$ oCIs, $N = 2$ gerbils) or in sacrificed animals ($n = 6$, $N = 6$). In wild-type animals, fewer multi-units (mainly located in dorsal ICC regions, i.e., tonotopically not corresponding to the illuminated regions in the basal cochlea) were responsive to light in hearing control gerbils. Responses had longer latencies (5.0 vs. 3.25 ms) and shorter duration (5.5 vs. 14 ms) as compared to neural responses in *CatCh*-injected animals (Fig EV4), consistent with the synaptic delay between the inner hair cell and SGNs upon opto-acoustic activation of the auditory pathway. Furthermore, responses were not stable over time and not

reproducible, as they vanished after the first recordings, potentially reflecting degradation of the endocochlear potential upon surgical manipulation of the cochlea.

Besides increasing response strengths of individual multi-units, recruitment of additional μLEDs also activated broader regions of the ICC: While individual μLEDs drove activity on an average of 13.0 ± 9.6 (out of a maximum of 32 electrodes, limited by the silicon probe design) electrodes per animal, blockwise stimulation evoked activity on 18.0 ± 8.6 electrodes, and oCI stimulation with all μLEDs recruited 24.12 ± 5.5 units (Fig 2J). This broadening of responses likely reflects the broader illumination of the spiral ganglion with an increased radiant flux. To demonstrate that these spatially restricted activation patterns originate from confined illumination by the oCI (both in space and in intensity) and not, for example, from spatially limited opsin expression in the spiral ganglion, we employed SGN stimulation with an optical fiber in a less selective manner in the same animals (Wrobel *et al*, 2018; Dieter *et al*, 2019): The fiber was placed to broadly illuminate the whole cochlea rather than confining laser light to a given cochlear region, and the intensity was set to ~35 mW, the highest possible laser output and by far exceeding the thresholds for *CatCh*-mediated SGN activation. Stimulating with these settings, multi-unit activity was observed at 30.3 ± 2.86 (out of 32) recording sites per animal, indicating that almost all units could potentially be driven by light. Thus, the selective activation of SGNs by oCI indeed originates from limited spread of light rather than from inhomogeneous SGN transduction (Fig 2J). The number of recruited multi-units did not differ between hearing and deafened AAV-injected animals upon oCI stimulation with blocks of μLEDs, all μLEDs, or optical fibers (Wilcoxon rank sum test, $P = 0.11, 0.14, 1$). However, a difference was found for stimulation with individual μLEDs of oCIs (Wilcoxon rank sum test, $P = 0.01$), which most likely results from the relatively low number of deafened animals used in this study. Nonetheless, as the average number of activated electrodes was higher in deafened animals (15.3 ± 10.2 vs. 11.7 ± 8.9;

**Figure 2. μLED-based oCIs evoke neural responses in the ICC.**

A–D    Exemplar STCs in response to SGN illumination with a single μLED (A), a block of four neighboring μLEDs (B), and all μLEDs of an oCI inserted via the round window in a *CatCh*-transduced gerbil (C) and with all μLEDs in a wild-type gerbil (WT; D). μLEDs were spaced by 250 μm.

E    Acoustic auditory brainstem responses (average of 1,000 stimuli at a stimulation rate of 10 Hz) upon click stimulation from 20 to 100 dB SPL before (left) and after deafening (right; star: threshold).

F    Optically evoked auditory brainstem responses before (left) and after deafening (right), elicited by ~35 mW pulses of 1 ms duration, delivered by a laser-coupled fiber (1,000 stimuli presented at 10 Hz), recorded from the same animal. Optogenetic activation of the auditory system was still possible after deafening. Differences in amplitude and waveform before and after deafening are likely due to different positioning of recording electrodes and the optical fiber.

G, H    Exemplar STCs in response to SGN illumination with all μLEDs in a *CatCh*-transduced, kanamycin-deafened gerbil before (G) and after (H) sacrifice.

I    Maximum strength of ICC responses (mean ± SD) evoked by oCI stimulation with 1 (indiv.; $n = 129/77$ μLEDs in $N = 9/3$ hearing/deaf gerbils), 4 (block; $n = 62/27$ blocks in $N = 9/3$ hearing/deaf gerbils), and 16 (all; $n = 24/9$ oCIs in $N = 9/3$ hearing/deaf gerbils) active μLEDs, and with a laser-coupled optical fiber (~35 mW; $n = 9/3$ fiber stimulations in $N = 9/3$ hearing/deaf gerbils) in *CatCh*-transduced gerbils (one-way ANOVA (F-value: 73.25, 4 degrees of freedom, $P = 5.19 \times 10^{-45}$) and post-hoc multiple comparison test; indiv. vs. block: $P = 9.9 \times 10^{-9}$; block vs. all: $P = 0.0096$; all vs. fiber: $P = 0.0016$; all *CatCh* vs. all WT: $P = 9.9 \times 10^{-9}$; **$P < 0.01$, ***$P < 0.001$, n.s. = non-significant). oCI stimulation with all μLEDs has also been performed in WT animals ($n = 7/7$ oCIs, $N = 2/2$ hearing/deaf gerbils) and in sacrificed *CatCh*-transduced animals (post-mortem; $n = 5/1$ oCIs in $N = 5/1$ hearing/deaf gerbils). Brown dots indicate data recorded from deafened animals.

J    Number of active electrodes (d′ > 1; mean ± SD) upon oCI stimulation with 1 ($n = 129/77$ μLEDs in $N = 9/3$ hearing/deaf gerbils), 4 ($n = 62/27$ blocks in $N = 9/3$ hearing/deaf gerbils), and 16 μLEDs ($n = 24/9$ oCIs in $N = 9/3$ hearing/deaf gerbils), and with optical fiber stimulation ($n = 9/3$ fiber stimulations in $N = 9/3$ hearing/deaf gerbils) in *CatCh*-transduced gerbils (one-way ANOVA (F-value: 30.18, 4 degrees of freedom, $P = 1.42 \times 10^{-21}$) and post-hoc multiple comparison test; indiv. vs. block: $P = 6.8 \times 10^{-5}$; block vs. all: $P = 0.006$; all vs. fiber: $P = 0.25$; all *CatCh* vs. all WT: $P = 9.9 \times 10^{-9}$; **$P < 0.01$, ***$P < 0.001$, n.s. = non-significant). Responses to oCI stimulation with all μLEDs were also observed in hearing WT animals ($n = 3$ oCIs in 2 gerbils). No active electrodes were found in response to 4/7 oCI stimulations in 2/2 hearing/deaf wild-type gerbils.

K    Thresholds (d′ = 1, measured at the best electrode (BE)) for ICC activation evoked with 16 active μLEDs on an oCI ($n = 24/9$ in $N = 9/3$ hearing/deaf gerbils) and with a laser-coupled optical fiber ($n = 9/3$ in $N = 9/3$ hearing/deaf gerbils; $P = 0.012$, Wilcoxon rank sum test). Central horizontal line indicates the median, lower, upper horizontal lines indicate the minimum and maximum values, and box edges indicate the 25th and 75th percentile, respectively.

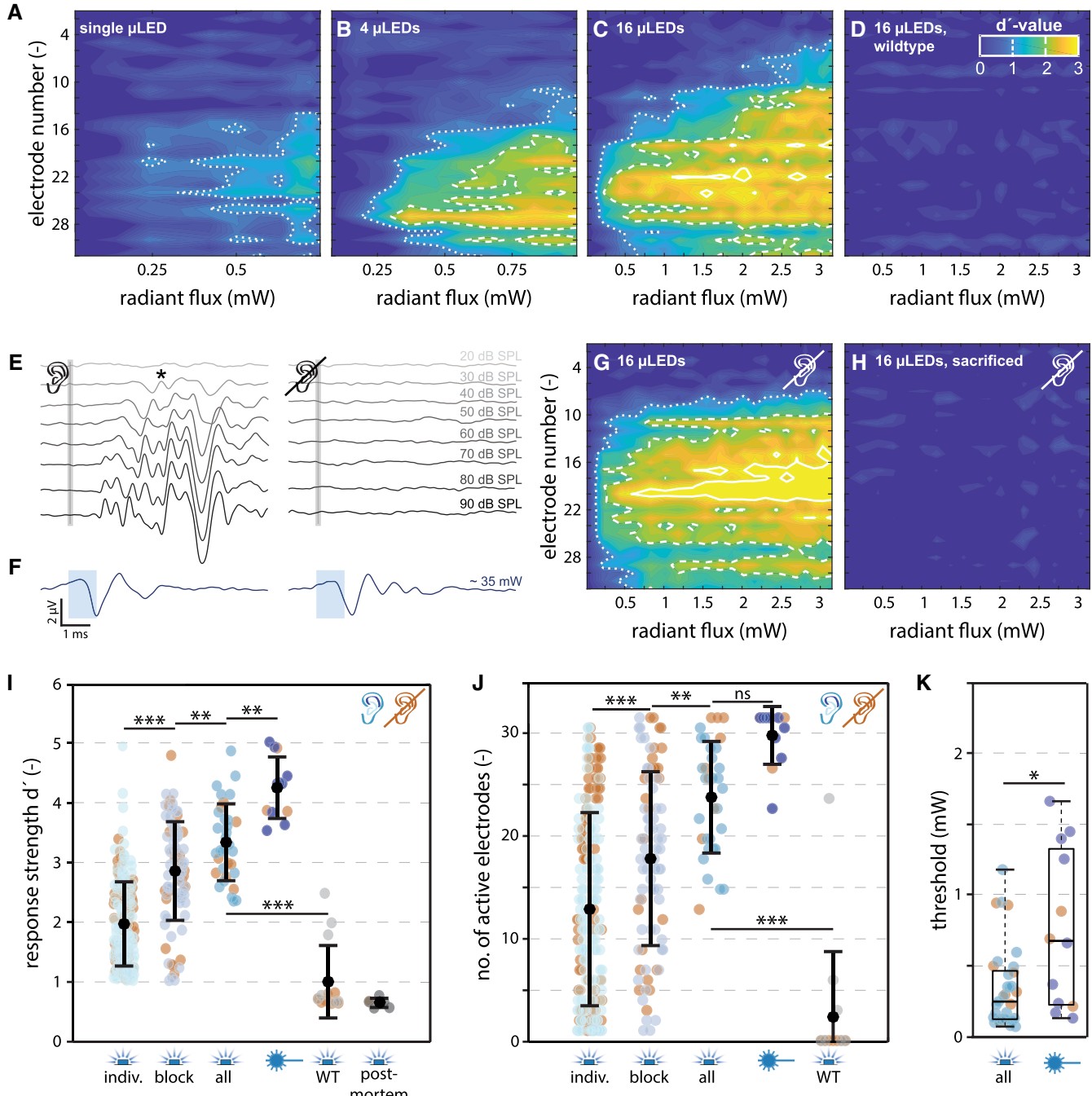

**Figure 2.**

mean ± SD; Wilcoxon rank sum test, $P = 0.01$) we can rule out that neural activation due to an opto-acoustic effect majorly contributed to the evoked responses recorded in hearing animals. Thresholds for optical stimulation of IC activity were lower for SGN illumination with oCIs (when all 16 μLEDs were driven) as compared to optical fibers (0.35 ± 0.25 mW vs. 0.76 ± 0.56 mW; $P = 0.015$, Wilcoxon rank sum test; Fig 2K), probably due to closer proximity of the emitter in respect to the SGNs. On average, a total driving current of 3.1 mA applied to all 16 μLEDs was required to exceed the threshold for optical activation.

## SGN activation with high frequency selectivity

In order to estimate activation patterns in the cochlea, the spatial extent of ICC activity has been quantified by an activity-based analysis. From STCs, the iso-contour line corresponding to a d' of 1 was taken as the threshold of neural activation, as previously done in other studies on electrical and optical auditory nerve stimulation (Snyder *et al*, 2004; Middlebrooks & Snyder, 2007; Richter *et al*, 2011; Dieter *et al*, 2019). The recording site with the lowest threshold, i.e., the focus of neural

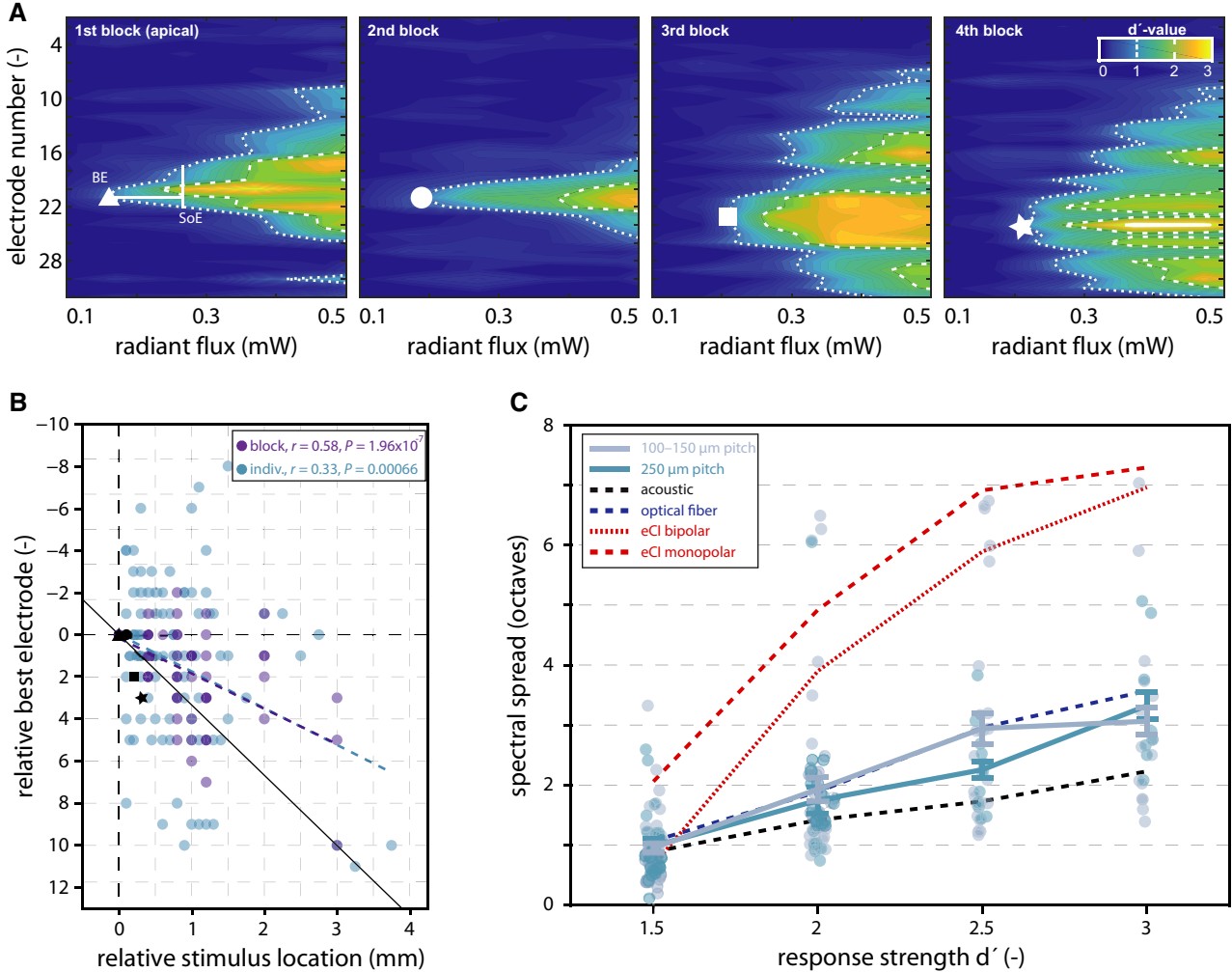

**Figure 3. Spectral features of oCI coding.**

A Exemplar STCs in response to SGN stimulation with four blocks of four μLEDs inserted via the round window, from apical to basal direction (A–D). White markers define the best electrode (BE), and measurement of the spread of excitation (SoE) is indicated in the left panel. In STCs where more than one peak was detected, the dorsal- and ventral-most electrodes have been considered as the boundaries of the STC in order to avoid underestimation of the SoE.

B Relative BE as a function of stimulus location, referring to the apical-most μLED evoking a response. The black line indicates the tonotopic axis as expected based on literature values, dashed lines indicate the regression line of individual (blue; correlation coefficient $r = 0.33$, $P = 0.00066$) and blockwise (purple; correlation coefficient $r = 0.58$, $P = 1.96 \times 10^{-7}$) oCI stimulation. Black markers indicate the data shown in panel A. Negative values can appear if the STC in response to a more basal emitter as compared to the apical-most emitter have a more dorsal BE.

C Spectral spread of excitation upon SGN stimulation with μLED-based oCIs (blocks of 4 μLEDs, solid lines; mean ± SEM), acoustic (black dashed line), laser-coupled optical fibers (blue dashed line), as well as mono- and bipolar electrical stimulation with a clinical-style eCI (dashed and dotted red lines, respectively). Average data from acoustic stimulation and stimulation via optical fibers and eCIs are replotted from Dieter et al (2019).

activation, was defined as the best electrode (BE). The spatial extent of neural activation, i.e., the spread of cochlear excitation, was defined as the range in the ICC covered by electrodes that recorded optically responsive multi-units at the stimulus intensity at which responses of a fixed d′ value were obtained from the BE (Fig 3A). This activity-based quantification allows for an estimation of neural excitation independent of the nature and absolute intensity of a stimulus, and thus allows for comparison across different modes of stimulation at comparable activation strengths. Since neural responses did not differ between hearing and deafened animals, the data of both groups were pooled for the subsequent analysis.

To demonstrate tonotopic activation of the auditory nerve by oCIs, SGNs have been stimulated both with individual μLEDs and with blocks of four neighboring μLEDs. The 16 μLEDs on oCIs were spaced either by 100, 150, or 250 μm, and thus covered a cochlear length of 1.5, 2.25, or 3.75 mm, respectively. Since the length of the scala tympani (where oCIs have been inserted) amounts to approximately 11 mm in gerbils (Dong & Olson, 2009), a maximum of 34.1% of the cochlear length could be covered by the oCIs. To overcome this limitation and demonstrate tonotopic activation over larger parts of the cochlea, some implants have been inserted via a cochleostomy in the middle cochlear turn rather than via the round window. Indeed, SGN illumination at distinct locations along the

cochlear spiral led to spatially confined neural activation of tono-topically corresponding ICC regions (Fig 3A).

To correct for different locations of oCI insertion (round window vs. cochleostomy), different insertion depths (due to varying implant lengths), and different μLED pitch, μLED locations in the cochlea were normalized to the apical-most μLED on each implant which elicited neuronal responses in the ICC. Hence, the BE of each STC has been normalized to the BE of the STC elicited by the apical-most μLED in the cochlea. In other words, for each implant and animal, μLED-dependent shifts in neural activation have been normalized to the apical-most μLED in the cochlea and the focus of neural activation evoked by this μLED. Upon SGN illumination with different μLEDs, a stimulus location-dependent shift of ICC activation was observed which amounted to 1.75 electrodes (i.e., 87.5 μm) in the ICC per millimeter stimulus location in the cochlea when stimulating with individual μLEDs (Pearson's $r = 0.33$, $P = 0.00066$) and 1.67 electrodes (i.e., 83.5 μm) when stimulating with blocks of four neighboring μLEDs (Pearson's $r = 0.58$, $P = 1.96 \times 10^{-7}$; Fig 3B). Hence, tonotopic activation of SGNs is achievable with μLED-based multi-channel oCIs.

Intracochlear spread of excitation (SoE) was estimated at different levels of response strength in the ICC. Since the average strength of neural responses to optical stimulation with individual μLEDs did not exceed a d' of 2 (Fig 2B) and neural activation could not be evoked in a reliable manner, blockwise stimulation with four neighboring μLEDs at a time was performed to estimate the spatial extent across a larger range of activation levels. Generally, the spread of neural excitation in the ICC upon intracochlear oCI stimulation increased with growing response strengths at the BE (Fig 3C). The spatial extent of neural excitation was then transferred to spectral spread by using the tonotopic slope of each animal (in case of deaf animals, this transformation was done by using the median tonotopic slope of hearing animals). The spread of excitation was similar when stimulating μLED blocks with a μLED pitch of 100–150 μm ($0.95 \pm 0.75/1.93 \pm 0.20/2.94 \pm 0.26/3.06 \pm 0.23$ octaves at d' of 1.5/2/2.5/3, respectively; mean ± SEM) compared to a pitch of 250 μm ($0.98 \pm 0.11/1.75 \pm 0.22/2.26 \pm 0.14/3.32 \pm 0.22$ octaves at d' of 1.5/2/2.5/3; $P = 0.81/0.60/0.22/0.69$, repeated-measures ANOVA ($F$-values: 0.06/0.28/1.52/0.16, respectively; $P = 0.81/0.60/0.25/0.69$, respectively, 1 degree of freedom) and post hoc pairwise comparison; Fig 3C). These findings were identical when the spread of ICC activation was not normalized by the tonotopic slope, which is reported as the physical space in the ICC (Appendix Fig S3).

## Discussion

In this study, we have applied multi-channel, μLED-based optical cochlear implants to activate the optogenetically modified auditory nerve, successfully combining the biomedical and optoelectronic work toward optogenetic hearing restoration. We demonstrated in vivo functionality in a tonotopic manner and with near-physiological spectral selectivity of optogenetic SGN stimulation by μLED-based oCIs in a rodent model of cochlear optogenetics.

### oCI implantation

For two of the implants, X-ray tomograms have been reconstructed (Fig 1J, Appendix Fig S4). Interestingly, light-emitting surfaces of

the μLEDs were facing Rosenthal's canal in both cases and hence achieved direct illumination of the spiral ganglion—a fact of critical importance for optogenetic hearing restoration. However, in both cases, the implant pierced through the basilar membrane at the base of the cochlea, crossing from the scala tympani to the scala vestibuli. This might be due to the limited flexibility of the implant: Both the planar structure of the μLEDs and the parallel power lines to address μLEDs limit the oCI flexibility to the plane of emitter surfaces. We speculate that the current oCIs were not flexible enough to be maintained in the scala tympani of the gerbil, respecting the basilar membrane. To overcome this problem, flexibility of the oCIs should be enhanced or oCIs should be implanted via a basal cochleostomy (such as shown in Fig 1I) to optimize the insertion angle with respect to the tangential axis of the basilar membrane in future studies.

### oCI-mediated neural activation

The average threshold of neural activation in the ICC upon oCI stimulation with all 16 μLEDs of SGNs was found to be 0.35 mW (0.35 μJ optical energy per pulse, 3.1 mA driving current, i.e., 3.1 μJ electrical energy per pulse). The maximum strength of neural responses in the ICC amounted to a d' of 1.97, 2.86, and 3.35 for oCI stimulation via 1 (activation-efficient), 4, and 16 active μLEDs, respectively. This compares to a d' of 4.27 upon fiber-based stimulation in the same animals and a maximum d' of 4.4 reported in a previous study, when driving SGNs with an optical power of up to 30 mW (Dieter et al, 2019). μLED stimulation reached ~75% of the response strength evoked by optical fibers, with an order of magnitude lower optical power (3.15 mW vs. 30 mW). However, optimization to increase light output of μLEDs is needed in order to fully utilize the dynamic range of neural firing, especially when stimulating with individual μLEDs. Generally, a high variability of responses was observed both for the maximum response strength and the number of activated μLEDs upon oCI activation (as demonstrated by the relatively high SD in Fig 2I and J). This variability might be attributed to different patterns of opsin expression across animals, the different layouts of oCIs used in this study (which differed in pitch and in total length, as well as in the patterns of functional/dysfunctional μLEDs on each implant), and the exact position of oCI implantation (cochleostomies vs. round window implantation). This variability could be overcome in the future by optimizing opsin expression, oCI stability, and implantation of oCIs. Furthermore, even if evoked activity differs across subjects, this might be counteracted to some degree by individualized cochlear implant fitting.

### Spectral selectivity of cochlear optogenetics

The average spatial spread of excitation in the ICC upon SGN stimulation with blocks of four neighboring μLEDs with a pitch of 100–150 μm amounted to 0.95/1.93/2.94/3.06 octaves at a d' of 1.5/2/2.5/3 at the BE, respectively. A previous study reported a spatial spread of 1.07/1.89/2.96/3.57 octaves upon confined fiber-based optogenetic stimulation, supporting our finding of spectrally precise neural activation of the auditory nerve (Dieter et al, 2019). While it might seem surprising that the spread of excitation with a block of 4 μLEDs, i.e., Lambertian emitters, is comparable to that of a 200 μm optical fiber, we suspect that this reflects the closer

proximity of μLEDs to the target tissue. Both optogenetic stimulation studies report a spread of excitation that comes—at least at low stimulation intensities—close to the one observed upon pure tone acoustic stimulation in non-injected control animals, which amounted to 0.89/1.42/1.73/2.23 octaves at d′ of 1.5/2/2.5/3. In contrast, the spread of excitation upon mono- and bipolar stimulation in the same study amounted to 2.06/4.92/6.91/7.29 and 0.67/3.90/5.89/6.96 octaves (Dieter *et al*, 2019). Thus, oCI-mediated activation of the auditory system is feasible with higher spectral selectivity than when using clinical-style eCIs, where even bipolar stimulation was less selective for all, except near threshold, intensities. Since we were only able to assess the spatial selectivity upon SGN stimulation with groups of four μLEDs, the spread of excitation upon stimulation with individual emitters still remains to be elucidated once light efficacy of μLED has been increased and/or opsins conveying a higher neural light sensitivity have been implemented. We expect that the spatial precision of optogenetic SGN stimulation will be higher when employing individual μLEDs and further improved when shaping the beam profile by conical concentrators and micro-lenses (Klein *et al*, 2019b).

### Non-optogenetic neural activation

Upon oCI stimulation in non-injected wild-type gerbils, weak neural responses were observed in 3 out of 8 implants. These responses differed from the optogenetically evoked neural responses in *CatCh*-transduced animals in the latency (Fig EV4A) and tonotopic position (Fig EV4B). We suggest that an opto-acoustic effect (Baumhoff *et al*, 2019) has evoked these responses, as the longer latency might result from the delay of synaptic transmission between the inner hair cell and the SGN, and responses could not be evoked in non-injected animals without hair cells. Furthermore, these responses vanished after some time and were not stable over the course of the experiment, most likely due to degradation of the endocochlear potential upon oCI implantation. In contrast, optically evoked responses in *CatCh*-transduced animals were highly stable over the whole duration of the experiments. However, even if there was a minor contribution to neural excitation by an opto-acoustic effect in the case of *CatCh*-transduced gerbils (which is unlikely given the comparable patterns of activation in regular hearing and deafened *CatCh*-injected animals), we have rather over-than underestimated the spread of excitation in this study, since such combined SGN excitation was most likely broader than for sole optogenetic stimulation as the optically evoked responses in wild-type gerbils where found more dorsally (lower frequency) in the ICC.

### Limitations

This study is only contributing one step in the effort to develop optical cochlear implants, which depends both on the optimization of an (opto)gene therapy and the oCI as a medical device. Toward the gene therapeutic aspect, specific and efficient expression of opsins within the tonotopic range of the spiral ganglion in adult animals remains a challenging task. In this study, robust oABRs could only be evoked in 43% of injected animals, which agrees with the success rate of 46% reported in an earlier study (Wrobel *et al*, 2018). In an oABR-negative animal in this study, only very sparse expression of *CatCh* has been observed (Fig EV2), which is

consistent with the previous finding that ~10% SGNs need to be transduced to evoke reliable oABRs (Wrobel *et al*, 2018). Future studies should thus focus on increasing the success rate of viral transduction, as well as the amount of transduced SGNs, to optimally render the spiral ganglion light-sensitive. Both this limited opsin expression, but also limited power output of individual μLEDs, might have led to the fact that neural responses could not be evoked with each functional μLED and evoked responses were far away from saturation (Fig 2B, individual μLEDs vs. optical fiber). Hence, one major objective of future μLED development is to increase the power output, which might be achieved by the implementation of micro-lenses (Klein *et al*, 2019b). Besides the power efficacy, the number of optical emitters and their pitch should as well be optimized for the model system used in future experiments. oCIs in this study housed 16 μLEDs with a maximum pitch of 250 μm, covering at most 3.75 mm, i.e., one-third, of the gerbil cochlear length. To optimally access the tonotopic axis of the auditory nerve with high precision, the number of emitters—ideally spaced by a pitch of 100 μm—should be increased. Furthermore, the depth of implantation in this study was limited mainly by the implant length, which should also be increased for future experiments in order to cover larger regions of the cochlea. A tapered oCI tip might further increase the implantation depth in order to maximize the coverage of cochlear tonotopy. Finally, biocompatibility and long-term stability of optical cochlear implants need to be evaluated in longitudinal studies upon chronic implantation, ideally by performing behavioral studies involving more complex stimulation patterns in order to determine the perceptual frequency resolution limit of optical sound encoding.

## Materials and Methods

### Animals

Data were obtained from 39 Mongolian gerbils (*Meriones unguiculatus*) of either sex between 6 and 14 months of age. Animals originated from our own breeding colony and were housed in a 12/12 h light–dark cycle with access to food and water *ad libitum*. Out of these 39 animals, 4 were non-injected control animals (2 hearing and 2 deafened) and 35 were *CatCh*-injected animals. Of these 35 animals, 20 were oABR negative and 15 oABR positive. Of the 15 oABR-positive animals, IC data could be recorded from 12 animals (9 hearing and 3 deafened). For each surgical procedure, gerbils were placed on a heating pad and anesthetized with isoflurane (induction: 4% at 1 l/min; maintenance: 1–2% at 0.4 l/min). Anesthetic depth was monitored by the absence of hind limb withdrawal reflexes and adjusted if necessary. Analgesia was achieved by injections of buprenorphine (0.1 mg/kg BW s.c.). All experiments were performed in compliance with the German national animal care guidelines and approved by the animal welfare office (LAVES; 17/2394) of the state of Lower Saxony, Germany, and the local animal welfare committee of the University Medical Center Göttingen.

### Gene transfer to SGNs

Virus suspensions of the recently engineered AAV-variant PHP.B (Deverman *et al*, 2016) carrying plasmids that code for the calcium-

permeable Channelrhodopsin-2-variant *CatCh* (Kleinlogel *et al*, 2011; $6.99 \times 10^{12}$ genome copies/ml, measured with qPCR AAV titration kit by TaKaRa/Clontech, according to the manufacturer's instructions; Keppeler *et al*, 2018), linked to the reporter eYFP, under control of the human synapsin promoter, have been injected into the spiral ganglion of adult Mongolian gerbils (> 3 months of age), as described before (Wrobel *et al*, 2018; Dieter *et al*, 2019). In summary, access to the cochlea was achieved via a retroauricular approach, and a small hole was drilled into the basal modiolus with a dental file to access the spiral ganglion (Chen *et al*, 2012; Wrobel *et al*, 2018; Dieter *et al*, 2019). After intramodiolar pressure injections of 2–3 μl virus suspension ($1.4–2.1 \times 10^{10}$ genome copies) via micropipettes, muscles and connective tissue were repositioned to close the surgical site, and the skin was sutured. Animals recovered for at least 4 weeks after surgery before electrophysiological measurements were performed.

### Deafening

Adult, opsin-injected animals were deafened by bilateral, intra-cochlear injections of kanamycin solution (~3 μl, 100 mg/ml, Kanamysel, Selectavet) 3–10 days before experiments were performed as described before (Wrobel *et al*, 2018). Briefly, the cochlea was accessed by the retroauricular approach described for virus injections, kanamycin solution was injected via the round window membrane, and a kanamycin-soaked gelatin sponge was placed in the round window niche before closing the surgical site with connective tissue and suturing the skin.

### Implant fabrication

Multi-channel oCIs were fabricated as described elsewhere (Klein *et al*, 2018, 2019a). Briefly, 16 μLEDs based on gallium nitride (GaN) with a footprint of $60 \times 60$ μm and a pitch of 100, 150, or 250 μm were integrated in a flexible, highly transparent, and biocompatible epoxy substrate. The μLEDs are individually addressable via a $4 \times 4$ matrix interconnection with 4 n- and 4 p-contacts. The μLED n-side contact was realized via a circular aperture (50 μm diameter) in the n-metallization on top of the emitting μLED surface. The well-established wafer-level process enables a high process yield (Klein *et al*, 2019a). In summary, μLEDs were fabricated from GaN epitaxially grown on sapphire substrates. The p-side contacts to the GaN were realized by a highly reflective stack of nickel, gold, and silver layers. The μLEDs were laterally defined by photolithography and subsequent chlorine-based plasma etching. The polymeric oCI substrates were realized via spin-coating of an epoxy resin on sapphire wafers equipped with a sacrificial aluminum (Al) layer. GaN-on-sapphire μLED and polymeric substrate wafers were subsequently joined via indium-gold inter-diffusion bonding. For improved mechanical stability, the gap between both wafers was underfilled with the same epoxy resin as used as substrate material. The μLED was released from the sapphire wafer by laser lift-off. Once the sapphire was removed, the emitting side of the μLED was exposed and roughened by a KOH etching process, compatible with the used epoxy (Fig 1F). By surface roughening, the light extraction was improved by up to 70%, dependent on the process duration of wet etching. The n-side contact and interconnecting lines were realized with a titanium–gold-based metallization insulated by an additional layer of epoxy. Finally, the stack of three polymeric layers was cut down to the sapphire carrier wafer using oxygen plasma to separate individual probes. Finally, probes were released by electro-chemical dissolution of the sacrificial Al layer and encapsulated in a layer of medical-grade silicone.

### Auditory brainstem recordings (ABRs)

Acoustic or optogenetic stimulation of the auditory pathway was verified by recordings of compound activity from the auditory nerve and brainstem in response to the stimulation (see next section for details, also see ref. Wrobel *et al*, 2018). Evoked potentials were recorded via subdermal, low-impedance needle electrodes at the vertex and the mastoid bone, amplified with a custom-made amplifier, and stored on a hard drive at a sampling rate of 50 kHz for offline analysis. A third needle placed at the back of the animals served as an active shielding electrode. Acoustic clicks (0.3 ms duration) or laser pulses (473 nm, 1 ms duration, delivered via a 200 μm optical fiber placed in the round window) of varying intensity have been presented at a rate of 10 Hz. Data were filtered between 300 and 3,000 kHz and averaged over 500–1,000 trials.

### Auditory midbrain recordings

Activity of multi-neuronal clusters (multi-unit activity, MUA) was recorded with linear 32-channel silicon probes (electrode area 177 μm², 1–3 MΩ impedance, 50 μm pitch; Neuronexus, Ann Arbor, US) from the ICC and was described in detail before (Dieter *et al*, 2019). Briefly, the ICC was stereotactically accessed via a craniotomy contralateral to the stimulated ear, and the silicon probe was inserted ~2 mm lateral and ~0.5 mm caudal to lambda to an initial depth of ~3.3 mm (measured from the surface of visual cortex, which partially covers the auditory midbrain in gerbils; Cant & Benson, 2005) using a micromanipulator (LN Junior 4 RE, Luigs & Neumann; Ratingen, Germany). After initial mapping of multi-unit activity with acoustic tones, the silicon probe was repositioned as needed in order to optimally access the tonotopic axis of the ICC (Ryan *et al*, 1982; Schnupp *et al*, 2015). An epidural low-impedance metal wire (< 1 Ω) served as a reference electrode on the contralateral hemisphere. Using the Digital Lynx 4s System (Neuralynx; Dublin, Ireland), multi-unit activity was amplified, filtered (0.1–9,000 Hz), digitized (32 kHz sampling rate), and stored on a hard drive for offline analysis. Once the preparation was done, stimuli were designed with custom-written MATLAB scripts (The MathWorks, Inc.; Natick, US) and generated with a custom-made system based on NI-DAQ-Cards (NI PCI-6229; National Instruments; Austin, US). Near-field acoustic stimulation was performed with a loudspeaker (Scanspeak Ultrasound; Avisoft Bioacoustics, Glienicke, Germany) positioned 30 cm in front of the animal's head and calibrated with a 0.25 inch microphone (4039; Brüel & Kjaer GmbH, Naerum, Denmark), pre-amplifier (2670), and measurement amplifier (2610). For optical SGN stimulation, access to the inner ear was realized with the retroauricular approach described for virus injections (Chen *et al*, 2012; Wrobel *et al*, 2018; Dieter *et al*, 2019). μLED-oCIs were inserted into the scala tympani via the round window or via cochleostomies in the basal or middle turn of the cochlea, with μLEDs facing the center of the cochlea. μLEDs were then driven individually or blockwise with current pulses of 1 ms

between 0 and 10 mA delivered (where the driving current of 10 mA split between the four µLEDs of one block, i.e., driving each µLED with ~2.5 mA) from a custom-made optical stimulator consisting of a microcontroller and a LED driver. For fiber-based stimulation, a laser-coupled (473 nm, 100 mW DPSS; Changchun New Industry Optoelectronics, Jilin, China) optical fiber (200 µm diameter, 0.39 NA; Thorlabs, Dachau, Germany) was inserted into the cochlea via the round window and directed toward the apex to broadly illuminate the cochlea. Data of electrical stimulation have been reprinted from Dieter and colleagues (Dieter *et al*, 2019). Briefly, a 4-channel clinical-style rodent CI (Wiegner *et al*, 2016) manufactured by MED-EL (Innsbruck, Austria) was inserted into the cochlea via the round window and biphasic pulses of 100 µs phase duration and varying intensity were delivered using a custom-made current source. An external ball electrode served as a return for monopolar electrical stimulation, whereas the electrode next to the stimulation electrode (in basal direction) served as a return for bipolar stimulation.

## Data analysis

All data analysis was performed with custom-written MATLAB scripts (R2019b). Time stamps of multi-units were extracted as peaks exceeding a threshold (median plus three median absolute deviations) from filtered data traces (0.6–6 kHz, 4th order Butterworth filter). After each time stamp, an artificial refractory period of 1 ms was implemented to avoid overestimating the spike rates. Frequency tuning of multi-units was assessed by presenting 100 ms pure tones (5 ms sine ramps) of frequencies between 0.5 and 32 kHz (quarter octave steps) and varying sound pressure level in a pseudo-random order at a repetition rate of 4 Hz. Frequency response areas were constructed from 20 to 30 repetitions of each frequency–intensity combination, and the frequency that evoked neural activity at the lowest intensity was defined as the characteristic frequency (Kiang *et al*, 1977; Egorova *et al*, 2001). For optical stimulation, multi-unit responses occurring 3–20 ms after stimulus onset (1 ms light pulses of varying intensity) were sorted into a response matrix according to stimulus intensity and recording site. Spatial tuning curves were then constructed based on the cumulative discrimination index of response rates in response to increasing stimulus intensity, as described previously (Snyder *et al*, 2004; Middlebrooks & Snyder, 2007; Richter *et al*, 2011; Dieter *et al*, 2019). Iso-contour lines were subsequently calculated by the *contour* function provided by MATLAB. The iso-contour line at a $d'$ of 1 was then defined as the threshold for neural activation, and the recording site with the lowest threshold was defined as the BE. The spread of neural excitation, i.e., the distance spanned by electrodes that recorded a $d'$ greater or equal to 1, was then calculated at the stimulus intensity that evoked a $d'$ of 1.5/2/2.5 or 3 at the BE. If electrodes which recorded responsive multi-units were separated by electrodes with multi-units below threshold ($d' = 1$), the distance between the dorsal- and ventral-most electrodes with a $d'$ greater or equal to 1 was used to calculate the spread of excitation in order to avoid underestimation. Data from oCIs in which STCs evoked by at least two different optical emitters reached threshold at a minimum of 3 electrodes and a $d'$ of 1.5 or more were taken to analyze tonotopic activation of the auditory nerve.

### The paper explained

**Problem**

Electrical cochlear implants restore hearing by electrically stimulating the auditory nerve. As the electric current spreads far from each electrode, precision of electrical hearing restoration is limited, resulting in poor frequency resolution of cochlear implants.

**Results**

Here, we optogenetically rendered the auditory nerve of the gerbil light sensitive and evoked neural responses with µLED-based, 16-channel optical cochlear implants. We demonstrated that the combination of gene therapy and microsystems engineering enables optical activation of the auditory nerve with higher spectral precision.

**Impact**

These results suggest that optogenetic stimulation of the auditory nerve with µLEDs might enable hearing restoration with improved frequency resolution to patients suffering from sensorineural hearing loss.

## Immunohistochemistry

Sample preparation and immunohistochemical staining were performed as described previously (Wrobel *et al*, 2018). Injected cochleae of a subset of animals were explanted after sacrificing the animals and fixed in 4% paraformaldehyde in PBS for 1 h at room temperature. Cochleae were decalcified in ethylenediaminetetraacetic acid (EDTA; 0.12 M for 3–4 days) and subsequently cryosectioned (16 µm thickness). Sections were blocked with goat serum dilution buffer (GSDB) for 1 h at room temperature, before primary antibodies for parvalbumin (1:300, guinea pig, cat. #195004, Synaptic Systems, Göttingen, Germany) and GFP (to label *CatCh*-EYFP, 1:500, chicken, cat. #ab13970, Abcam, Cambridge, United Kingdom) were incubated at 4°C overnight. Goat anti-guinea-pig-568 (1:500 in GSDB, cat. #A11075, Thermo Fisher Scientific, Waltham, US) and goat anti-chicken-488 (1:200 in GSDB, cat. #A11039, MoBiTec, Göttingen, Germany) were used as secondary antibodies and incubated at room temperature for 1 h. Images were acquired using an Abberior Instruments Expert Line microscope in confocal mode using a 20× oil immersion objective or on a Zeiss LSM 510 using a 40× air objective. Images were processed for display in ImageJ.

## X-ray tomography

After acquisition of electrophysiological data, some oCIs have been fixed into the scala tympani with dental cement. Positioning of the implant and direction of the µLEDs light-emitting surface was assessed with X-ray tomography as described previously (Bartels *et al*, 2013; Wrobel *et al*, 2018). Data acquisition was achieved with a customized imaging system for cone-beam in-line phase-contrast tomography based on a liquid metal X-ray source and a LuAG scintillator-based detector with a pixel size of 6.5 µm, and a fast Fourier-based phase reconstruction procedure. Segmentation and visualization of reconstructed structures were achieved with the Avizo 3D 9 software, and cochlear structures as well as oCI components were traced semi-automatically.

**Statistics**

Pearson's correlation has been used to quantify the correlation between auditory or optogenetic stimuli and the tonotopic position of evoked responses. The Wilcoxon rank sum test, as a non-parametric test, has been used to quantify differences between two independent groups, i.e., recorded in hearing and deafened animals, respectively. We want to note caution for these comparisons, as datasets recorded from deafened animals are smaller (both in terms of tested oCIs and of the number of animals). As this study was limited by the number of available implants, we have decided to perform most experiments in CatCh-injected, hearing animals, in order to quantify the response properties. In deafened animals, we mainly intended to demonstrate the feasibility of optogenetic SGN stimulation by μLEDs in an animal model of sensorineural hearing loss, rather than quantifying these measures and comparing them against hearing animals. Analysis of variance has been used to compare more than two groups of data.

## Data and software availability

The MATLAB code used to analyze the data published in this study is provided as source data. Source data and analysis code can also be downloaded from http://www.innerearlab.uni-goettingen.de/materials.html.

**Expanded View** for this article is available online.

### Acknowledgements
The authors are very grateful to Daniela Gerke for expert help with virus preparation. We thank Maria Michael for providing the histology of the oABR-negative gerbil (Fig EV2) and Jakob Neef for help with the graphical abstract. We thank Ben Deverman and Viviana Gradinaru for providing the PHP.B construct used in this study. We thank Peter Wenig and Daniel Weihmüller for expert technical support on electronics. The work was funded by the European Research Council (ERC) under the European Union's Horizon 2020 research and innovation program (Grant agreement No. 670759—advanced grant "OptoHear") to TM, the grant OpticalCI of the German Ministry of Research and Education (no. 13N13729) to PR and TM, and supported by the Deutsche Forschungsgemeinschaft (DFG, German Research Foundation) under Germany's Excellence Strategy—EXC 2067/1-390729940 to TM as well as support of MED-EL to PR and TM. AD is a fellow of the German Academic Scholarship Foundation.

### Author contributions
AD, EK, MJ, PR, and TM designed the study. EK and PR designed optical cochlear implants with input of AD, DK, MJ, and TM. EK manufactured optical cochlear implants under the supervision of PR and OP. LJ and TH designed software for driving optical cochlear implants. GH designed hardware for driving optical cochlear implants. MJ designed hardware and software for multi-channel electrophysiological recordings. VR produced the virus used in this study and performed immunohistochemistry of SGNs. AD performed virus injections and electrophysiological recordings. AD performed data analysis under the supervision of MJ and TM. DK performed post-mortem X-ray reconstructions and initial insertion studies in explanted cochleae to optimize oCI design. AD and TM prepared the initial draft of the manuscript. All authors edited and finalized the manuscript.

### Conflict of interest
TM and DK are co-founders of the company OptoGenTech.

### For more information
(i)  https://www.who.int/news-room/fact-sheets/detail/deafness-and-hearing-loss
(ii)  http://eurociu.eu/
(iii)  https://www.asha.org/public/hearing/Cochlear-Implant/
(iv)  http://www.auditory-neuroscience.uni-goettingen.de/hearing_the_light_EN.html

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
