## [Review Process File · EMBO Molecular Medicine]

μLED-based optical cochlear implants for spectrally selective activation of the auditory nerve

Alexander Dieter, Eric Klein, Daniel Keppeler, Lukasz Jablonski, Tamas Harczos, Gerhard Hoch, Vladan Rankovic, Oliver Paul, Marcus Jeschke, Patrick Ruther, and Tobias Moser
DOI: [10.15252/emmm.202012387](https://doi.org/10.15252/emmm.202012387)

Corresponding author(s): Tobias Moser (tmoser@gwdg.de) , Patrick Ruther (ruther@imtek.de)

Review Timeline:

Submission Date:	24th Mar 20
Editorial Decision:	27th Apr 20
Revision Received:	23rd May 20
Accepted:	2nd Jun 20

Editor: Celine Carret

Transaction Report:

27th Apr 2020

Dear Prof. Moser,

Thank you for the submission of your manuscript to EMBO Molecular Medicine. We have now heard back from the three referees whom we asked to evaluate your manuscript.

You will see from the set of comments pasted below, that the three referees are very positive about the study and only request minor revisions. Therefore, I would like to ask you to perform these minor revisions and, in order to gain time, to also consider the below editorial amendments before submitting your next version:

1) Please address the minor comments from the three referees.

Provide a point-by-point letter INCLUDING my comments as well as the reviewer's reports and your detailed responses to their comments (as Word file).

2) Please carefully check the authors guidelines for formatting your supplemental information:

Expanded view and Appendix (see:

<https://www.embopress.org/page/journal/17574684/authorguide#expandedview>).

An Appendix pdf file should be submitted separately, including all the supplementary information.

Please note the different nomenclature and content (i.e. supplementary methods should be moved into the main article), update call outs.

3) Figures: separate figure files should be provided

4) In the main manuscript file, please do the following:

- correct/answer the track changes suggested by our data editors and myself by working from the attached document

- add up to 5 keywords

- change "Methods" for "Materials and Methods"

- ethics: provide the gerbils gender, handling conditions and origin of purchase. Age at which they were used should be indicated in the article as well.

- in M&M, the statistical paragraph should reflect all information that you have filled in the Authors checklist, especially regarding randomisation, blinding, replication.

- indicate in legends exact $n=$ and exact $p=$ values, not a range, along with the statistical test used.

Some people found that to keep the figures clear, providing an Appendix table Sx with all exact p -values was preferable. You are welcome to do this if you want to.

5) EMBO Molecular Medicine now requires a complete author checklist

(<http://embomolmed.embopress.org/authorguide#editorial3>) to be submitted with all revised

manuscripts. Please use the checklist as guideline for the sort of information we need WITHIN the manuscript. This is particularly important for animal reporting, the use of human samples, antibody dilutions and exact p - and n -values that should be indicated instead of a range, along with the right justified statistical test.

This file will be published along the Review Process File.

6) For more information: There is space at the end of each article to list relevant web links for further consultation by our readers. Could you identify some relevant ones and provide such information as well? Some examples are patient associations, relevant databases, OMIM/proteins/genes links, author's websites, etc...

7) The Paper Explained: EMBO Molecular Medicine articles are accompanied by a summary of the articles to emphasize the major findings in the paper and their medical implications for the non-specialist reader. Please provide a draft summary of your article highlighting

- the medical issue you are addressing, = Problem
- the results obtained = Results
- their clinical impact = Impact

8) Every published paper now includes a 'Synopsis' to further enhance discoverability. Synopses are displayed on the journal webpage and are freely accessible to all readers. They include a short stand first (maximum of 300 characters, including space) as well as 2-5 one sentence bullet points that summarise the paper. Please write the bullet points to summarise the key NEW findings. They should be designed to be complementary to the abstract - i.e. not repeat the same text. We encourage inclusion of key acronyms and quantitative information (maximum of 30 words / bullet point). Please use the passive voice. Please attach these in a separate file or send them by email, we will incorporate them accordingly.

You are also encouraged to suggest a striking image or visual abstract to illustrate your article. If you do please provide a jpeg file 550 px-wide x (250-400)-px high.

9) As part of the EMBO Publications transparent editorial process initiative (see our Editorial at <http://embomolmed.embopress.org/content/2/9/329>), EMBO Molecular Medicine will publish online a Review Process File (RPF) to accompany accepted manuscripts.

In the event of acceptance, this file will be published in conjunction with your paper and will include the anonymous referee reports, your point-by-point response and all pertinent correspondence relating to the manuscript. Let us know whether you agree with the publication of the RPF.

10) Data and software availability. Please add the following:

"Data and software availability" (within the M&M)

"The datasets and computer code produced in this study are available in the following databases:

- Modeling computer scripts: GitHub (<https://github.com/SysBioChalmers/GECKO/releases/tag/v1.0>)
- [data type]: [full name of the resource] [accession number/identifier] ([doi or URL or identifiers.org/DATABASE:ACCESSION])"

Please submit the MATLAB codes/scripts to repositories or provide them as source data within this submission, referenced in this section.

11) Dr. Ruther must obtain an ORCID number and associate it to this manuscript.

I look forward to seeing a revised form of your manuscript as soon as possible (however, given the exceptional current circumstances, we perfectly understand that delays are to be expected. If you don't mind, just update us after 3 months to let us know where you are standing with the revision).

Yours sincerely,

Celine Carret

Celine Carret, PhD
Senior Editor
EMBO Molecular Medicine

*** Instructions to submit your revised manuscript ***

**** PLEASE NOTE **** As part of the EMBO Publications transparent editorial process initiative (see our Editorial at <https://www.embopress.org/doi/pdf/10.1002/emmm.201000094>), EMBO Molecular Medicine will publish online a Review Process File to accompany accepted manuscripts.

To submit your manuscript, please follow this link:

Link Not Available

- 1) a .doc formatted version of the manuscript text (including Figure legends and tables). Please make sure that the changes are highlighted to be clearly visible to referees and editors alike.
- 2) separate figure files*
- 3) supplemental information as Expanded View and/or Appendix. Please carefully check the authors guidelines for formatting Expanded view and Appendix figures and tables at <https://www.embopress.org/page/journal/17574684/authorguide#expandedview>
- 4) a letter INCLUDING the reviewers' reports and your detailed responses to their comments (as Word file)

Also, and to save some time should your paper be accepted, please read below for additional information regarding some features of our research articles:

5) The paper explained: EMBO Molecular Medicine articles are accompanied by a summary of the articles to emphasize the major findings in the paper and their medical implications for the non-specialist reader. Please provide a draft summary of your article highlighting

6) For more information: There is space at the end of each article to list relevant web links for further consultation by our readers. Could you identify some relevant ones and provide such information as well? Some examples are patient associations, relevant databases, OMIM/proteins/genes links, author's websites, etc...

7) Author contributions: the contribution of every author must be detailed in a separate section (before the acknowledgments).

8) EMBO Molecular Medicine now requires a complete author checklist (<https://www.embopress.org/page/journal/17574684/authorguide>) to be submitted with all revised manuscripts. Please use the checklist as a guideline for the sort of information we need WITHIN the manuscript as well as in the checklist. This is particularly important for animal reporting, antibody dilutions (missing) and exact p-values and n that should be indicated instead of a range.

9) Every published paper now includes a 'Synopsis' to further enhance discoverability. Synopses are displayed on the journal webpage and are freely accessible to all readers. They include a short stand first (maximum of 300 characters, including space) as well as 2-5 one sentence bullet points that summarise the paper. Please write the bullet points to summarise the key NEW findings. They should be designed to be complementary to the abstract - i.e. not repeat the same text. We encourage inclusion of key acronyms and quantitative information (maximum of 30 words / bullet point). Please use the passive voice. Please attach these in a separate file or send them by email, we will incorporate them accordingly.

You are also welcome to suggest a striking image or visual abstract to illustrate your article. If you do please provide a jpeg file 550 px-wide x 400-px high.

10) A Conflict of Interest statement should be provided in the main text

11) Please note that we now mandate that all corresponding authors list an ORCID digital identifier. This takes <90 seconds to complete. We encourage all authors to supply an ORCID identifier, which will be linked to their name for unambiguous name identification.

Currently, our records indicate that the ORCID for your account is 0000-0001-7145-0533.

Link Not Available

12) The system will prompt you to fill in your funding and payment information. This will allow Wiley to send you a quote for the article processing charge (APC) in case of acceptance. This quote takes into account any reduction or fee waivers that you may be eligible for. Authors do not need to pay any fees before their manuscript is accepted and transferred to our publisher.

Photos 400-800 DPI

*Additional important information regarding figures and illustrations can be found at <http://bit.ly/EMBOPressFigurePreparationGuideline>

***** Reviewer's comments *****

Referee #1 (Remarks for Author):

This work by Dieter, Klein et al. describes the application of multi-channel μ LED optical cochlea implants (oCIs) to evoke neuronal activity in optogenetically modified spiral ganglion neurons (SGNs). The overall translational goal is to overcome the limitations of electrical cochlear implants for hearing restoration, for example current spread and limited spectral selectivity of SGNs activation. This manuscript represents an important proof of concept showing optically evoked tonotopic activity of the auditory pathway and spectral selectivity of optogenetic SGN stimulation by 16 μ LED-based oCIs. Of note, the biological (efficient optogenetic transduction of SGN), engineering (biocompatible and stable μ LED-based oCIs) and surgical (oCI transplantation) aspects are extremely challenging, require a precise interplay and adjustment and have been mastered by the authors. There are some minor points that need to be addressed and discussed in more detail.

Minor comments:

- 1) Please define "oABR" the first time it is mentioned, not every interested reader knows that optically-evoked auditory brainstem responses are meant.
- 2) oCIs may offer advantages over eCIs, however so far, "high" spectral selectivity could be misleading due to the fact that single μ LED stimulation was not successful and block-illumination was required (simultaneous shining with bunch of μ LEDs). It would be great to discuss this aspect in more detail in regard to the performances of eCIs.
- 3) Does the evoked ICC activity by individual or block μ LEDs follow particular patterns coupled to the optical stimulation? Are there any differences in activity depending on each individual μ LED?
- 4) Is there a distribution/standard deviation of the light emission power among single LEDs of an array with respect to the same applied current?

- 5) The emission characteristics such as wavelength/bandwidth of μ LEDs are not stated.
- 6) All signal profiles have been represented as an average of 1000 pulses. It would be great to show at least one example of a signal profile at baseline and during light stimulation.
- 7) How many SGN neurons are activated by one LED channel?
- 8) Why is the rate of opsin function after transfection in animals (12/32) so low? Please discuss/report the challenges for the general readers.

Referee #2 (Remarks for Author):

The manuscript " μ LED-based optical cochlear implants for spectrally selective activation of the auditory nerve" by Dieter et al is the latest advance from this research group in developing optical cochlear implants, which could have a profound impact on individuals with hearing loss. The authors use optogenetic tools that they previously developed to infect both hearing and deafened gerbils with a fast channelrhodopsin variant targeted to cochlear spiral ganglion neurons (SGN), which comprise the ascending auditory nerve. In previous work the channelrhodopsins were activated using optical fibers to excite the auditory nerve. In this work, the authors demonstrate a significant advance in the method of light delivery to the cochlea, using micro-LEDs in a multi-channel array. The array could be positioned in different locations in the cochlea based upon type of implantation surgery. Further, the authors demonstrate that optical stimulation of the SGN has a reduced "spectral spread", indicating improved tonotopic activation of the auditory nerve, compared to traditional electrical cochlear implants. Overall, the work is solid and polished. Some analyses are complex and difficult to understand (see a comment below), but figures are well done, and analyses are appropriate.

1. The function of the optical cochlear implants compared to electrical cochlear implants is well done. Please also discuss how the function relates to the spectral selectivity of sound activation of the auditory system in general, or IC neurons in particular for this study. Based on the first sentence on page 4, you have this data.
2. First sentence page 4 - were the acoustic stimulation experiments performed in the same animals as the optical stimulation? This is not clear. When was this performed in relation to oCI insertion?
3. Is it known why oABRs failed in 20/32 animals? Was this an expression problem?
4. In figure 1D', it would be helpful to label the IHC location to distinguish from the eYFP in the type I dendrite
5. Figure 2 and 3 should refer to the "example" or "exemplar" data, as "exemplary" is used to mean the best or perfect example
6. Figure 3B is a particularly difficult to understand figure, but there is results text dedicated to explaining it. Further improvements could be made by indicating which data points in 3B refer to the panels in 3A, and in clearly detailing why some data points are negative on the y-axis.

Referee #3 (Remarks for Author):

This study is addressing the feasibility of a new generation of a cochlear implant using optogenetics instead of electrical stimulation. The team of T.Moser is a leader in the field. They already published two major papers in the field (Science trans med, 2018 and nature comm, 2019). In this paper, they go further using a fibre equipped with 16 micro-LEDs to allow a spectral selective

activation of the nerve fibre. To assess this selectivity, they illuminate SGN at different locations and then recorded corresponding inferior colliculus regions. This study uses cutting edge technologies from various fields of research (AAV, electrophysiological recordings, optogenetics, etc.). The study is well constructed and nicely described and constitutes a crucial preclinical breakthrough that allows going closer to the clinics. Also, they are aware that this study is not perfect and that they need to improve the power output of the LEDs if they want to apply optogenetic CI to human.

One concern is the low number of animals used and the discordance in the size of the different groups (i.e. 9 controls and 3 deafened AAV injected gerbils) even if finally, it does not change the conclusions of the findings.

Material and methods should be carefully reviewed and more detailed to allow reproducibility of the experiments in the scientific community.

The titer of AAV: the authors should explain and justify the titer used (mentioned precisely what is GC, how they obtained 6.99×10^{12} GC/ml and why do they injected only a final concentration of $1.4 - 2.1 \times 10^{10}$ GC as in the previously cited studies they used much more (i.e. in Wrobel paper they injected $3.2 \times 10^{12} - 2.7 \times 10^{13}$ genome copies/ μ l

ABRs are not detailed at all and do not refer to any other published paper

Age of the animal for deafening?

- Discordance in the number of animals used: 16 gerbils are mentioned in the "mat and meth" section. In results, the authors said: "After verifying opsin function (robust oABRs could be evoked in 12/32 animals)". Could they explain the different numbers?

Figure 1J: yellow is not visible

Figure 2 All the example of STCs should be presented at the same scale for radiant flux (mainly to compare panels C and G that are strikingly different and should be explained).

Statistical analysis should be better described (for example the authors wrote: "As for response strengths, also the number of recruited multi-units did not differ between hearing and deafened AAV-injected animals ($p = 0.03/0.24/0.14/0.52$ for individual/block/all/fiber stimulation, Bonferroni-corrected, two-sample t-test, $p = 0.05/4 = 0.0125$ "). Is it counterintuitive to say that there is no difference if $p=0.03$ or 0.05 ? In addition, the authors should mention that they performed an ANOVA if this is the case and why did they perform t-test or any other test such as Wilcoxon ((figure 3 legend: "($n = 9/3$ in $N = 9/3$ hearing/deaf gerbils; $p = 0.58$, Wilcoxon rank sum test)"). To avoid a surcharge of the text, an additional table with all the statistics clearly explained should be performed.

Referee #1 (Remarks for Author):

This work by Dieter, Klein et al. describes the application of multi-channel μ LED optical cochlea implants (oCIs) to evoke neuronal activity in optogenetically modified spiral ganglion neurons (SGNs). The overall translational goal is to overcome the limitations of electrical cochlear implants for hearing restoration, for example current spread and limited spectral selectivity of SGNs activation. This manuscript represents an important proof of concept showing optically evoked tonotopic activity of the auditory pathway and spectral selectivity of optogenetic SGN stimulation by 16 μ LED-based oCIs. Of note, the biological (efficient optogenetic transduction of SGN), engineering (biocompatible and stable μ LED-based oCIs) and surgical (oCI transplantation) aspects are extremely challenging, require a precise interplay and adjustment and have been mastered by the authors. There are some minor points that need to be addressed and discussed in more detail.

We are grateful to Referee 1 for the appreciation of our work, as well as for the comments, which helped us to further improve our manuscript.

Minor comments:

1) Please define "oABR" the first time it is mentioned, not every interested reader knows that optically-evoked auditory brainstem responses are meant.

Thanks for pointing this out, we have added this information.

2) oCIs may offer advantages over eCIs, however so far, "high" spectral selectivity could be misleading due to the fact that single μ LED stimulation was not successful and block-illumination was required (simultaneous shining with bunch of μ LEDs). It would be great to discuss this aspect in more detail in regard to the performances of eCIs.

In response to this comment, we have avoided the claim of "high spectral selectivity" throughout the MS. Specifically, in most cases, we have followed the advice of the reviewer to discuss it in comparison to the eCI. The information on the spread of excitation with mono- and bipolar electrical stimulation and acoustic stimulation (taken from our previous study that used the same set-up for comparison to fiber-based optogenetic stimulation) is provided in the discussion and we have added the spread of excitation of acoustic stimulation to Fig. 3C, which supports the notion of "near physiological" spectral selectivity.

3) Does the evoked ICC activity by individual or block μ LEDs follow particular patterns coupled to the optical stimulation? Are there any differences in activity depending on each individual μ LED?

At this point, we do not have an analysis regarding the particular pattern of individual μ LEDs or μ LED blocks. However, we do observe substantial variability among the responses, probably due to different patterns of opsin expression, oCI-design (pitch, pattern of functional μ LEDs), and oCI position upon implantation (see also response to reviewer 3). We have commented on this variability now in the discussion.

4) Is there a distribution/standard deviation of the light emission power among single LEDs of an array with respect to the same applied current?

Characterizing a batch of 16 exemplary oCI samples with 16 μ LEDs each using optimal positioning toward the detector we extracted at a μ LED current of 10 mA applied at a pulse duration of 1 ms an averaged optical output power of 0.932 ± 0.010 mW (average \pm standard deviation (SD); $n = 256$), respectively. The SD of the individual oCIs varied between 0.6 and 2% of the respective oCI-averaged output power.

5) The emission characteristics such as wavelength/bandwidth of μ LEDs are not stated.

Done, 462 nm, we quote measurements in Klein et al., Front Neurosci 2018 (Figure 13 therein)

6) All signal profiles have been represented as an average of 1000 pulses. It would be great to show at least one example of a signal profile at baseline and during light stimulation.

The thousand repetitions are averaged for oABR data, which is correct. However, due to their small signal amplitude, no response can be observed in an individual trial. Multiunit activity-data – or the resulting d' values of them – originate from 30 repetitions only. We have now included an individual trace, as well as a corresponding scatter plot in response to 30 stimulus presentations, for a multi-unit recorded from the ICC in the supplementary figure S2.

7) How many SGN neurons are activated by one LED channel?

This is a great question, but difficult to answer. As indicated by the spread of excitation measurements shown for blocks of 4 μ LEDs in Fig. 3, this will heavily depend of the radiant flux of the μ LED. Assuming i) 24.000 type I SGNs (early postnatal count, (Richter *et al*, 2011), ii) a uniform distribution along the tonotopic axis (8.8 octaves (Müller, 1996), and iii) a fraction of 30% of SGNs expressing CatCh across all tonotopic places (Wrobel *et al*, 2018), we approximate that maximally ~800 SGNs (3% of total) are recruited for the lowest spread of excitation obtained with blocks of 4 μ LEDs (~ 1 octave at $d' = 1$).

8) Why is the rate of opsin function after transfection in animals (12/32) so low? Please discuss/report the challenges for the general readers.

Yes, the low fraction of opsin expressing SGNs upon intramodiolar AAV-injection into the adult gerbil cochlea is troublesome and we have extended the discussion of this point in response to the reviewer's comment. In brief, we suspect that AAV access to the SGN represents the major bottleneck. While we and others are actively working on improving AAV-mediated transduction of adult SGN e.g. by using more potent AAV vectors (e.g. AAV-PHP.B, this study) and application of larger volumes of AAV suspension to the scala tympani, we have not yet managed the reproducibility and transduction rates we find with early postnatal AAV injections into rodent cochleae (100% of surviving animals, 40-9X% of SGNs transduced). Within a previous study (Wrobel *et al*, Science Translational Medicine, 2018) we have extensively looked at the cochleae of oABR-negative animals, and could not find robust opsin expression. Unfortunately, we did not process the tissue of most of the negative animals in this study, but we have had a look at one of these cochleae, and have included this data to the MS now (Fig. EV2). While PV-staining revealed the presence of SGNs, the GFP-expression indicating *CatCh* was really sparse.

Referee #2 (Remarks for Author):

The manuscript "μLED-based optical cochlear implants for spectrally selective activation of the auditory nerve" by Dieter et al is the latest advance from this research group in developing optical cochlear implants, which could have a profound impact on individuals with hearing loss. The authors use optogenetic tools that they previously developed to infect both hearing and deafened gerbils with a fast channelrhodopsin variant targeted to cochlear spiral ganglion neurons (SGN), which comprise the ascending auditory nerve. In previous work the channelrhodopsins were activated using optical fibers to excite the auditory nerve. In this work, the authors demonstrate a significant advance in the method of light delivery to the cochlea, using micro-LEDs in a multi-channel array. The array could be positioned in different locations in the cochlea based upon type of implantation surgery. Further, the authors demonstrate that optical stimulation of the SGN has a reduced "spectral spread", indicating improved tonotopic activation of the auditory nerve, compared to traditional electrical cochlear implants. Overall, the work is solid and polished. Some analyses are complex and difficult to understand (see a comment below), but figures are well done, and analyses are appropriate.

We thank Referee 2 for the appreciation of our work, as well as for the comments, which helped us to further improve our manuscript.

1. The function of the optical cochlear implants compared to electrical cochlear implants is well done. Please also discuss how the function relates to the spectral selectivity of sound activation of the auditory system in general, or IC neurons in particular for this study. Based on the first sentence on page 4, you have this data.

It is true that we have acoustic data from these animals for confirming the recording position of the electrode array in the ICC. However, we do not think that comparison to this data is fair, as the mapping has been performed after cochlear surgery, and thus hearing in these animals is already compromised and not as good as in wildtype animals (as shown in Dieter et al, Nature Communications, 2019, thresholds across the hearing range of the gerbil are elevated by ≥ 20 dB in animals that underwent cochlear surgery). However, we have recorded data in response to pure tone acoustic stimulation in naïve gerbils in this previous study, and have now added the average spread of excitation obtained from this data to Fig. 3C of the current manuscript for ease of orientation.

2. First sentence page 4 - were the acoustic stimulation experiments performed in the same animals as the optical stimulation? This is not clear. When was this performed in relation to oCI insertion?

We have added a sentence to outline the experimental flow more clearly. Acoustic stimulation has been done in all non-deafened animals. It was, however, done after performing cochlear surgeries which most likely have compromised hearing (by a threshold shift of ~ 20 dB across the whole hearing range, as shown in Dieter et al, Nat. Comm. 2019) and is thus not comparable to natural hearing in a naïve animal.

3. Is it known why oABRs failed in 20/32 animals? Was this an expression problem?

Yes, the low fraction of opsin expressing SGNs upon intramodiolar AAV-injection into the adult gerbil cochlea is troublesome and we have extended the discussion of this point in response to the reviewer's comment. In brief, we suspect that AAV access to the SGN represents the major bottleneck. While we and others are actively working on improving AAV-mediated transduction of adult SGN e.g. by using more potent AAV vectors (e.g. AAV-PHP.B, this study) and application of larger volumes of AAV suspension to the scala tympani, we have not yet managed the reproducibility

and transduction rates we find with early postnatal AAV injections into rodent cochleae (100% of surviving animals, 40-9X% of SGNs transduced). Within a previous study (Wrobel et al, Science Translational Medicine, 2018) we have extensively looked at the cochleae of oABR-negative animals, and could not find robust opsin expression. Unfortunately, we did not process the tissue of most of the negative animals in this study, but we have had a look at one of these cochleae, and have included this data to the MS now (Fig. EV2). While PV-staining revealed the presence of SGNs, the GFP-expression indicating *CatCh* was really sparse.

4. In figure 1D', it would be helpful to label the IHC location to distinguish from the eYFP in the type I dendrite.

Done.

5. Figure 2 and 3 should refer to the "example" or "exemplar" data, as "exemplary" is used to mean the best or perfect example.

Thanks for pointing this out, we have changed the phrase as suggested.

6. Figure 3B is a particularly difficult to understand figure, but there is results text dedicated to explaining it. Further improvements could be made by indicating which data points in 3B refer to the panels in 3A, and in clearly detailing why some data points are negative on the y-axis.

Thanks for helping us to make these results more accessible. We have now indicated the data points for best electrodes in figure 3A in figure 3B. As before, there were examples in figure 3A which were taken from cochleostomy and round window implantation (i.e. two different experiments), we have now chosen an example of blockwise stimulation with one oCI. While this might not shift the BE as much as different oCI insertions, we can now refer to four data points from the same implant in figure 3B, and thereby hopefully contribute to a more intuitive understanding of this figure. Also, we have added a short sentence about the origin of negative data points.

Referee #3 (Remarks for Author):

This study is addressing the feasibility of a new generation of a cochlear implant using optogenetics instead of electrical stimulation. The team of T. Moser is a leader in the field. They already published two major papers in the field (Science trans med, 2018 and nature comm, 2019). In this paper, they go further using a fibre equipped with 16 micro-LEDs to allow a spectral selective activation of the nerve fibre. To assess this selectivity, they illuminate SGN at different locations and then recorded corresponding inferior colliculus regions. This study uses cutting edge technologies from various fields of research (AAV, electrophysiological recordings, optogenetics, etc..). The study is well constructed and nicely described and constitutes a crucial preclinical breakthrough that allows going closer to the clinics. Also, they are aware that this study is not perfect and that they need to improve the power output of the LEDs if they want to apply optogenetic CI to human.

One concern is the low number of animals used and the discordance in the size of the different groups (i.e. 9 controls and 3 deafened AAV injected gerbils) even if finally, it does not change the conclusions of the findings.

We thank Referee 3 for the appreciation of our work and the remarks, which helped us to further improve our manuscript. We want to note that the two different groups (9 hearing and 3 deafened animals) are not meant to be compared against each other. Rather, we have quantified results of the pooled data ($n = 12$), but wanted to demonstrate in a subset ($n = 3$) of animals, that our technology works in deafened animals as well. Unfortunately, we have used up all oCIs with this design for the current study, and oCIs of the next generation will have a different design (as mentioned in the discussion), which prevents us from doing additional experiments with the current settings.

Material and methods should be carefully reviewed and more detailed to allow reproducibility of the experiments in the scientific community.

Fair point, done and we have added the requested information.

The titer of AAV: the authors should explain and justify the titer used (mentioned precisely what is GC, how they obtained 6.99×10^{12} GC/ml and why do they injected only a final concentration of $1.4 - 2.1 \times 10^{10}$ GC as in the previously cited studies they used much more (i.e. in Wrobel paper they injected $3.2 \times 10^{12} - 2.7 \times 10^{13}$ genome copies/ μ l).

Thanks for catching this one!!

GC = genome copies, as this abbreviation is only used twice, we now spare this abbreviation. 6.99×10^{12} GC/ml were measured by the qPCR AAV titration kit by TaKaRa, as we now have added, and $1.4 - 2.1 \times 10^{10}$ does not refer to a concentration, but rather approximate the number of genome copies that have been injected, as calculated by the volume of the injected virus suspension (2-3 μ l). It is true that the viral titer differs from Wrobel et al. (2018), where we now realized that we have made a mistake. The reported titer there should be $3.2 \times 10^{12} - 2.7 \times 10^{13}$ genome copies/ml, which compares to the titer used in the current study, we are about to submit a correction for Wrobel et al. (2018).

ABRs are not detailed at all and do not refer to any other published paper.

We have included additional information and a reference to a previous paper with detailed description.

Age of the animal for deafening? Has been added.

Discordance in the number of animals used: 16 gerbils are mentioned in the "mat and meth" section. In results, the authors said: "After verifying opsin function (robust oABRs could be evoked in 12/32 animals)". Could they explain the different numbers?

We apologize for not stating this clearly. We could evoke oABRs in 12 out of 32 opsin-injected animals and performed IC recordings in these we animals (9 hearing and 3 deafened). In addition, we have added 4 non-injected wildtype animals (two hearing and two deafened), which then adds up to the 16 gerbils mentioned in the materials and methods section. Actually, to complicate it even more, we have had 3 more animals showing oABRs (which increase the fraction of positive animals in our study) for which, however, we could not obtain IC recordings, either due to death of the animal, or to misplacement of the electrode array. We have now added these animals to the total number, and explained the number of animals more clearly in general, hopefully avoiding confusion.

Figure 1J: yellow is not visible.

As this is not essential, we have removed the sentence, as the visibility of the encapsulated cables and μ LEDs might be compromised when using a more prominent color for the encapsulation.

Figure 2 All the example of STCs should be presented at the same scale for radiant flux (mainly to compare panels C and G that are strikingly different and should be explained).

We now present all the panels at the same radiant flux, which makes these figures more comparable. It is correct, that STCs in response to different oCIs and recorded from different animals are quite variable, which might result from different opsin expression patterns, different oCI-designs (in terms of pitch, and in terms of functional/unfunctional μ LEDs on individual devices), and different positions of implants. We now chose more comparable STCs to avoid confusion, but we added these points to the discussion.

Statistical analysis should be better described (for example the authors wrote: "As for response strengths, also the number of recruited multi-units did not differ between hearing and deafened AAV-injected animals ($p = 0.03/0.24/0.14/0.52$ for individual/block/all/fiber stimulation, Bonferroni-corrected, two-sample t-test, $p = 0.05/4 = 0.0125$ "). Is it counterintuitive to say that there is no difference if $p=0.03$ or 0.05 ? In addition, the authors should mention that they performed an ANOVA if this is the case and why did they perform t-test or any other test such as Wilcoxon ((figure 3 legend: "($n = 9/3$ in $N = 9/3$ hearing/deaf gerbils; $p = 0.58$, Wilcoxon rank sum test)"). To avoid a surcharge of the text, an additional table with all the statistics clearly explained should be performed. Thanks for pointing this out. We have discussed this, and adopted statistical tests to be more adequate (mainly Wilcoxon Rank Sum and ANOVA). We have stated the exact test in each figure legend, and have summarized the statistics in a supplementary table.

2nd Jun 2020

Dear Tobias,

Thank you very much for your reply and for providing the last set of items. We are pleased to inform you that your manuscript is accepted for publication and is now being sent to our publisher to be included in the next available issue of EMBO Molecular Medicine.

We would like to remind you that as part of the EMBO Publications transparent editorial process initiative, EMBO Molecular Medicine will publish a Review Process File online to accompany accepted manuscripts. If you do NOT want the file to be published or would like to exclude figures, please immediately inform the editorial office via e-mail.

Please read below for additional IMPORTANT information regarding your article, its publication and the production process.

Congratulations on your interesting work,

Celine Carret

Celine Carret, PhD
Senior Editor
EMBO Molecular Medicine

Follow us on Twitter @EmboMolMed
Sign up for eTOCs at embopress.org/alertsfeeds

*** ** IMPORTANT INFORMATION ** **

SPEED OF PUBLICATION

The journal aims for rapid publication of papers, using using the advance online publication "Early View" to expedite the process: A properly copy-edited and formatted version will be published as "Early View" after the proofs have been corrected. Please help the Editors and publisher avoid delays by providing e-mail address(es), telephone and fax numbers at which author(s) can be contacted.

Should you be planning a Press Release on your article, please get in contact with embomolmed@wiley.com as early as possible, in order to coordinate publication and release dates.

LICENSE AND PAYMENT:

All articles published in EMBO Molecular Medicine are fully open access: immediately and freely available to read, download and share.

EMBO Molecular Medicine charges an article processing charge (APC) to cover the publication costs. You, as the corresponding author for this manuscript, should have already received a quote with the article processing fee separately. Please let us know in case this quote has not been

received.

Once your article is at Wiley for editorial production you will receive an email from Wiley's Author Services system, which will ask you to log in and will present you with the publication license form for completion. Within the same system the publication fee can be paid by credit card, an invoice, pro forma invoice or purchase order can be requested.

Payment of the publication charge and the signed Open Access Agreement form must be received before the article can be published online.

PROOFS

You will receive the proofs by e-mail approximately 2 weeks after all relevant files have been sent to our Production Office. Please return them within 48 hours and if there should be any problems, please contact the production office at embopressproduction@wiley.com.

Please inform us if there is likely to be any difficulty in reaching you at the above address at that time. Failure to meet our deadlines may result in a delay of publication.

All further communications concerning your paper proofs should quote reference number EMM-2020-12387-V2 and be directed to the production office at embopressproduction@wiley.com.

Thank you,

Celine Carret, PhD
Senior Editor
EMBO Molecular Medicine

Corresponding Author Name:	Prof. Dr. Tobias Moser, Dr. Patrick Ruther
Journal Submitted to:	EMBO Molecular Medicine
Manuscript Number:	EMM-2020-12387-V2